# Heparanase-1: From Cancer Biology to a Future Antiviral Target

**DOI:** 10.3390/v15010237

**Published:** 2023-01-14

**Authors:** Nadjet Lebsir, Fabien Zoulim, Boyan Grigorov

**Affiliations:** 1Université Claude Bernard Lyon 1, INSERM 1052, CNRS 5286, Centre Léon Bérard, Centre de Recherche en Cancérologie de Lyon, 69434 Lyon, France; 2Confluence: Sciences et Humanités (EA 1598), UCLy, 10 Place des Archives, 69002 Lyon, France; 3Hospices Civils de Lyon, 69002 Lyon, France

**Keywords:** heparanase-1, cancer, virus infections, heparanase-1 inhibitors

## Abstract

Heparan sulfate proteoglycans (HSPGs) are a major constituent of the extracellular matrix (ECM) and are found to be implicated in viral infections, where they play a role in both cell entry and release for many viruses. The enzyme heparanase-1 is the only known endo-beta-D-glucuronidase capable of degrading heparan sulphate (HS) chains of HSPGs and is thus important for regulating ECM homeostasis. Heparanase-1 expression is tightly regulated as the uncontrolled cleavage of HS may result in abnormal cell activation and significant tissue damage. The overexpression of heparanase-1 correlates with pathological scenarios and is observed in different human malignancies, such as lymphoma, breast, colon, lung, and hepatocellular carcinomas. Interestingly, heparanase-1 has also been documented to be involved in numerous viral infections, e.g., HSV-1, HPV, DENV. Moreover, very recent reports have demonstrated a role of heparanase-1 in HCV and SARS-CoV-2 infections. Due to the undenied pro-carcinogenic role of heparanase-1, multiple inhibitors have been developed, some reaching phase II and III in clinical studies. However, the use of heparanase inhibitors as antivirals has not yet been proposed. If it can be assumed that heparanase-1 is implicated in numerous viral life cycles, its inhibition by specific heparanase-acting compounds should result in a blockage of viral infection. This review addresses the perspectives of using heparanase inhibitors, not only for cancer treatment, but also as antivirals. Eventually, the development of a novel class antivirals targeting a cellular protein could help to alleviate the resistance problems seen with some current antiretroviral therapies.

## 1. Introduction

Heparanase-1 is an endo-beta-D-glucuronidase of 50 kDa that cleaves the heparan sulphate (HS) side chains of the heparan sulphate proteoglycans (HSPGs). This results in the release of saccharide products of 4–7 kDa that can still associate with protein ligands (cytokines, growth factors, etc.), and facilitate their biological activity. Heparanase-1 is synthesized as a 65 kDa proenzyme in the Golgi and is further processed by cathepsin L in late endosomes into two subunits of 50 kDa and 8 kDa that form a proteolytically active enzyme. The latter is secreted in the extracellular matrix (ECM), where it exerts its HS-degrading function [1]. A second form of heparanase has also been reported, heparanase-2, but shows no HS-degrading activity [2]. Heparanase-1 has also been shown to possess signaling functions, regardless of its enzymatic activity [3], and may translocate into the nucleus and associate with the euchromatin, where gene transcription typically occurs, resulting in an increase in cellular proliferation [4]. As a side effects of the heparanase-1 hydrolytic activity of HSPG and ECM loosening, various pre-existing growth factors and cytokines associated with HSPG are released and act as angiogenesis and wound healing enhancers, a phenomena first described by Vlodavsky’s team [5]. The present review provides a summary of the physiopathology of heparanase-1 in cancer and viral infections with a particular focus on heparanase inhibition as a possible target for the development of novel antiviral strategies.

## 2. Implication of Heparanase-1 in Carcinogenesis

The expression of heparanase-1 is a finely regulated physiological process. Whereas heparanase-2 expression is widely spread in normal tissues, the expression of heparanase-1 is instead restricted to the placenta, keratinocytes, platelets, activated immune cells, and immune organs (bone marrow, spleen, lymph nodes) endothelial cells, with little expression in connective tissue cells and epithelia [6,7,8,9,10,11]. In normal tissues, the heparanase-1 promoter is inhibited by the promoter methylation and direct binding of wild type p53 [12]. During normal cellular processes, heparanase-1 expression can be upregulated in response to various stimuli, e.g., upon immune cell activation or placenta formation [13,14,15,16]. The dysregulation of heparanase-1 expression in cancer has been extensively documented over the two past decades and forms a key hallmark of cancer (Figure 1).

There is ample evidence to indicate a correlation between heparanase-1 activity and the metastatic potential of tumor cells [10,17,18,19,20]. Moreover, heparanase-1 overexpression has long been reported in various solid tumors and is associated with poor prognosis in bladder [21], gastric [22], colorectal [23], liver [24], cervical [25], and breast cancers compared to matched normal tissues [26]. The upregulation of heparanase-1 expression has also been documented in hematological malignancies, e.g., for multiple myeloma, in which heparanase-1 represents a prominent biomarker for its early detection and a promising therapeutic target [27,28,29,30,31]. The overexpression of heparanase-1 has been identified as an important mediator of the processes tightly associated with tumorigenesis, such as ECM reorganization, cell proliferation, migration, epithelial to mesenchymal transition (EMT), angiogenesis, and the promotion of tolerant immune responses within the tumor microenvironment [32].

### 2.1. Primary Tumor Formation

The most fundamental feature in primary tumor formation implies the ability of cancer cells to sustain chronic proliferation and evade growth suppressors. Proto-oncogenes, such as Myc and Ras, are positive cell-cycle regulators that control a wide range of gene programs involved in cell growth, proliferation, and differentiation. The aberrant activation of these proteins has a major impact on a variety of cellular pathways and represents the most frequent oncogenic alterations in human cancers that appear at early stages in tumorigenesis [33,34]. For example, it has been reported that heparanase-1 cooperates with Ras to drive malignant development in breast and skin tissue in a murine model by inducing an over-proliferation of epithelial cells and acinar structure disorganization [35]. Furthermore, this appears to be a two-way cooperation process as heparanase-1, in turn, is upregulated by oncogenes. This was reported for B-Raf kinase, which exerts protumorigenic properties via heparanase promoter activation [36]. In addition, a positive correlation between c-myc and heparanase-1 was observed in gastric cancer through the indirect upregulation of heparanase-1 via the activation of human telomerase reverse transcriptase (hTERT) [37]. The involvement of heparanase-1 in such crucial steps of primary tumor initiation and in close cooperation with key proto-oncogenes, as well as tumor suppressors, places the enzyme among the key pro-tumoral factors. Indeed, although heparanase-1 has no role in inducing mutations in the *TP53* tumor suppressor gene, its expression is upregulated by mutant p53 variants, whereas wild-type p53 binds to the heparanase promoter and inhibits its activity [12]. Recently, heparanase-1 was also reported to decrease the activity of the potent tumor suppressor PTEN activity by remodeling chromatin conformation and regulating its stability in the nucleus [38].

### 2.2. Tumor Microenvironment Modulation

Heparanase-1 enzyme activity plays a prominent role in tumor microenvironment modulation. Within the ECM, the HS chains sequester a broad set of cytokines and growth factors, known as HS binding proteins (HSBP), thereby limiting their bioavailability in the surrounding microenvironment. Exacerbated remodeling of the ECM by heparanase-1 is observed in the early stages in cancer (tumorigenesis), but also in late phases (i.e., tumor invasion and metastasis). During tumor initiation, a pleiotropic factor belonging to HSBP, the fibroblast growth factor (FGF), exerts its pro-tumorigenic functions via autocrine and paracrine loops by modulating cell proliferation, survival, differentiation, migration, and metabolism [39]. In fact, heparanase-1 enzymatic activity contributes to the tumor-promoting functions of the FGF through HS degradation and, under certain circumstances, through HS sulfation [40,41]. Indeed, the latter was found to be associated with various human cancers [42]. Heparanase-1 knock-down in a proximal tubular cell line prevents the epithelial-mesenchymal transition (EMT) induced by FGF-2 [43].

### 2.3. Neo-Angiogenesis Induction

In the same way as for normal tissues, tumors require nutrients and oxygen for sustenance, as well as a means of eliminating metabolic waste. Such a fundamental need for cancer cells to survive is possible through the process of tumor-associated neoangiogenesis. The vascular endothelial growth factor (VEGF) is an important player in angiogenesis and vascular permeability features among the key HSBP. Mounting evidence points to the involvement of VEGF and/or its receptor in the growth of primary tumors and metastasis, where they are often found to be highly expressed [44,45]. Heparanase-1 induces VEGF expression through the activation of the Arc, Akt, and p38 signaling pathways, independent of its enzymatic activity. This results in enhanced angiogenesis and lymphangiogenesis [46,47]. In the same manner, the activation of the Src pathway via Rac-1, a small GTPase, efficiently stimulates cell adhesion and spreading through the clustering of cell surface HS proteoglycans.

### 2.4. Immune Response Modulation and Fibrosis Enhancement

Several other mechanisms that tie heparanase-1 with cancer progression have been described, as is the case for the involvement of heparanase-1 in immune response modulation [48], enhanced exosome formation in tumor cells [49], as well as in autophagy leading to chemotherapy resistance [50]. Heparanase-1 also has a role in inflammation by facilitating the recruitment of immune cells in the context of cancer development, as is the case for the tumor-associated macrophages (TAM) that play an adverse role in pancreatic adenocarcinoma pathogenesis [48]. Indeed, TAM are known to produce bioactive molecules, such as cytokines, growth factors, and antiapoptotic molecules, which support the tumor microenvironment [51]. Heparanase-1 is also involved in fibrogenesis as a consequence of inflammation, by regulating TGF-β, as it has been shown for renal fibrosis [52].

### 2.5. Metastasis

Following its characterization, it did not take very long to realize that heparanase-1 was not only involved in normal HS catabolism within the ECM, but that it was also implicated in cancer and, more precisely, metastasis [5,6]. The first connection between heparanase-1 and metastasis was provided by Nakajima and his colleagues [5]. Indeed, the metastatic potential of highly invasive melanoma cells to extravasate and successfully colonize the lung was related to the capacity of heparanase-1 to degrade HS in the walls of pulmonary blood vessels [5]. Similarly, the same conclusion was made for lymphoma cells: by solubilizing the glycosaminoglycan (GAG) scaffolding of the sub-endothelial ECM, heparanase-1 facilitates the spread and extravasation of metastatic T lymphoma cells through the blood vessel walls [6]. This cell extravasation-promoting effect of heparanase-1 is not restricted to cancer cells. Indeed, heparanase-1 has been described to help leukocytes to traverse blood vessel walls on their way to infection sites [7,8]. The extravasation of leukocytes is followed by diapedesis throughout the endothelium, through which they come across the basement membrane and the underlying connective tissue, made up mostly of interlacing collagen fibrils bundled with flexible proteoglycans, such as HSPG [9]. The degradation of these linear polysaccharides is an important step to achieve the transmigration of leukocytes. However, cancer cells can also benefit from the heparanase-1 activity for extravasation and metastasis processes [10].

These studies emphasize the role of heparanase-1 in mediating the crosstalk between tumor cells and the tumor microenvironment. Some other experimental studies have even tried to benefit from the ECM degradation capacity of heparanase-1 by engineering tumor-specific CAR (chimeric antigen receptor) expressing heparanase-1, which are capable of infiltrating stroma-rich solid tumors and of specifically killing cancer cells [53].

## 3. Heparanase-1 Inhibitors for Anti-Cancer Therapy

Heparanase-1 has an important role in multiple cancers. Aberrant heparanase-1 expression is correlated with inflammation, tumor angiogenesis, metastasis, tumor aggressiveness, and poor prognosis. Experiments with different cancer cell lines and tumor models (e.g., lymphoma, melanoma, urothelial cells), in which heparanase-1 expression was downregulated by RNA interference, showed reduced invasion or adhesion, and increased apoptosis [54,55]. Therefore, due to the undeniable pro-cancerogenic role of heparanase-1, multiple treatment strategies have been developed to inhibit its enzymatic activity. The utility of developing such inhibitors is strengthen by the following facts: (i) there is only one form of enzymatically active heparanase (heparanase-1) [1,2]; (ii) its expression is quite limited in healthy tissues [6,7,8,9,10,11]; (iii) heparanase-knocked-out animals showed no particular deficiencies or phenotypes [56]. These facts imply that heparanase inhibitors will have only minimal side effects in clinics and make heparanase-1 an important target for anti-cancer therapy. The 3D structure of heparanase-1 revealed that sulfation is key for the interaction of heparanase with HS, and that the recognized cleavage site is a trisaccharide accommodated into the heparanase binding cleft [57]. This can suggest multiple interactions with structurally diverse HS polysaccharide substrates [58]. Hence, the main heparanase inhibitors that have been developed are HS mimetics and act by competition.

Heparin, a sulfated glycosamynoglycan, being the closest mimic of HS, has been initially considered as a putative heparanase-1 inhibitor, but its strong anticoagulant activity limits its use as an anti-cancer treatment [59]. Therefore, considerable efforts have been made in the development of modified heparins and related polysulfated compounds with reduced anticoagulant activity, such as modified heparins (N-acetylated, glycol-split heparin) or sulfated oligosaccharides (e.g., PI-88, PG545, roneparstat) [59,60]. Roneparstat (SST0001) is a 15–25 kDa N-acetylated and glycol split heparin with low anticoagulant activity and high anti-heparanase-1 activity with a potential anti-tumor role [29,61]. However, a phase I clinical study revealed little efficacy of roneparstat in multiple myeloma, although it has a safe and well tolerated profile [29]. Pixatimod (PG545) is a synthetic HS mimetic and a strong heparanase-1 inhibitor with anti-cancer properties (inhibitor of angiogenesis, tumor growth and metastasis). This compound shows mild anticoagulant activity, a common side effect associated with HS mimetics [62]. PGP545 has shown significant activity in breast, colon, pancreatic, lung, and ovarian tumor models with a tolerable safety profile, as documented in a phase I study [63]. Among all heparanase-1 inhibitors, muparfostat (PI-88) is the only one to have passed phase II and III clinical studies so far, and has good safety and tolerability profiles in patients [64,65,66,67]. PI-88 is a HS mimetic. It is a mixture of chemically sulfated oligosaccharides derived from the yeast *Pichia (Hanensula) holstii* Y-2488, ranging between di- to hexa-saccharides, with the major components being penta-saccharide (60%) and tetra-saccharide (30%) [60,65]. In addition to acting as a heparanase-1 inhibitor, PI-88 blocks angiogenesis directly by antagonizing the interactions of angiogenic growth factors, such as FGF-2 and VEGF, and their receptors, with HS [68]. The results from the phase III study in hepatocellular carcinoma patients after surgical resection showed that PI-88 could significantly prolong disease-free survival in subgroup patients with microvascular invasion; however, this was not the case in the overall treatment group, in which no significant improvement was observed.

In parallel, selective small molecule heparanase inhibitors have been designed and synthesized. These include benzoxazoles [69], benzoimidazoles [70], HS-configured pseudodisaccharides [58], and heparanase-neutralizing antibodies [71], all of which showed specific anti-heparanase-1 and anti-cancer activity in vitro in nanomolar concentrations. Some of them, such as cyclophellitol-derived pseudodisaccharides, can irreversibly inhibit heparanse-1 by covalently binding to the enzyme active site and have shown metastasis reduction in animal models [58]. The search for small synthetic molecules is justified by their ease of production, the possibility of oral administration and their improved pharmacokinetics profiles. For more information on these inhibitors, refer to [72].

## 4. Heparanase-1 Involvement In Virus Infections and Their Inhibition

Regardless of the undeniable role of heparanase-1 in cancer development and progression, many emerging studies emphasize its involvement in viral infection and pathogenesis. Accumulating evidence indicates a role of heparanase-1 in the lifecycle of many viruses, including herpes simplex virus 1 and 2 (HSV-1, HSV-2), human papillomavirus (HPV), dengue virus (DENV), hepatitis C virus (HCV), porcine reproductive and respiratory syndrome virus (PRRSV), respiratory syncytial virus (RSV), and, recently, the severe acute respiratory syndrome, coronavirus 2 (SARS-CoV-2) (Table 1).

In these reports, the authors often used gene downregulation approaches to study the impact of heparanase-1 deficiency on virus infection or exogenously overexpressed heparanase-1. For example, HSV-1 infection was significantly reduced in the epithelium of the cornea through the shRNA targeting of the heparanse-1 gene [73,74]. Moreover, the transfection of non-functional heparanase-1 mutants in these cells inhibited viral release. In another report, in which heparanase-1 was shown to promote HSV-2 release, the overexpression of heparanse-1 clearly increased the infection and virus release from vaginal epithelial VK2 cells [75]. Another example is the study by Guo et al., in which they investigated the role of heparanase-1 in porcine reproductive and respiratory syndrome virus (PPRSV) infection [78]. For this purpose, they both downregulated heparanase-1 expression by siRNAs and overexpressed it through the transfection of cells with a heparanase-1 overexpressing plasmid. This led to the suppression of PRRSV replication and egress in the first case, whilst an enhanced virus release was observed in the second. Similar genetic approaches have recently shown the implication of heparanase-1 in hepatitis C virus (HCV) infection [79].

It can therefore be assumed that the inhibition of heparanase-1 by specific heparanase-acting compounds should result in a block of viral infection. In this section, we summarize the major findings in virus-heparanase-1 interactions, as well as discussing the possible targeting of the enzyme by specific compounds in order to block infection (Figure 2).

### 4.1. Herpes Simplex Virus 1 and 2 (HSV-1, HSV-2)

The involvement of heparanase-1 in a virus lifecycle was initially documented for HSV-1 and HSV-2. Herpes is a lifelong pathology, is mostly asymptomatic, and is characterized by periodic viral replication. The last available epidemiological report, from 2016 and by the World Health Organization (WHO), indicates a prevalence for HSV-1 and for HSV-2, of 67% and 13%, respectively, in the worldwide population [82,83]. HSV-1 replication is limited to the epithelia, commonly causing an infection in or around the mouth, in the eye cornea, or the genital area, and can also set latency in the sensory neurons [84,85]. Its transmission is mostly oral-to-oral, whereas HSV-2 is primarily sexually transmitted and is the primary cause of genital herpes [85].

As is the case with many viruses, HSV-1 needs to attach to HS on the cell surface to permit receptor-mediated cell entry involving several other entry factors, e.g., HVEM (herpesvirus entry mediator), Nectin-1, and Nectin-2. HS interacts with the HSV-1 envelope glycoproteins, gB and gC, during the initial attachment step in HSV-1 entry [86]. However, during the viral egress steps, HS may form an obstacle for virion release by tethering the newly made virions to the plasma membrane. To alleviate this budding obstacle, HSV-1 induces high heparanase-1 expression in the infected cells, resulting in enhanced HS cleavage and virus release. Further analyses shows that HSV-1 enhanced heparanase-1 expression in a NF-κB dependent manner [74,87]. The fine regulation of HS by heparanase-1 during the HSV-1 infection cycle plays an important role in maintaining a balance between HSV-1 entry and viral release. Other reports also support the substantial link between heparanase-1 and the pathologies associated with HSV-1. In human corneal epithelial cells, heparanase-1 is upregulated upon infection by HSV-1 via the viral protein ICPγ34.5, which is involved in viral replication and egress [74] and leads to efficient viral spread. In HSV-1 infected cells, heparanase-1 translocates to the nucleus to increase the production of pro-inflammatory factors and cytokines (IL-1β, IL-6, TNF-α), which in turn leads to tissue destruction and impaired wound healing [73]. These results confirm the role of heparanase-1 as a driver of HSV-1 viral pathogenesis, which is also valid for HSV-2.

The pathogenesis of vaginal herpes by HSV-2, another member of the HSV family, is also linked to the role of heparanase-1 in virion release. The inhibition of heparanase-1 strongly reduces progeny virus release from infected epithelial cells [75]. During HSV-2 infection, heparanase-1 expression is upregulated by NF-κB and again facilitates virus release through the removal of HS from the cell surface.

Different heparanase-acting compounds have been used when studying HSV infection. In human corneal epithelial cells (HCE), the pharmacological inhibition of heparanase-1 by the benzoxazole inhibitor OGT2115 blocked the spread of HSV-1 from infected to uninfected cells. Moreover, the virus release from these cells was completely abolished by using OGT2115 at 10 µM [73]. The same inhibitor at the same concentration also blocked HSV-2 egress and spread in human vaginal epithelial cells [75]. Interestingly, pixatimod (PG545), a cholestanol-conjugated sulfated oligosaccharide, exhibited virucidal activity against HSV-2 particles in vitro, manifested as a disruption of their lipid envelope [88], as well as in vivo, as this compound, protected mice against genital infection with HSV-2.

### 4.2. Human Papillomavirus (HPV)

HPV is another DNA virus that hijacks heparanase-1 for its own benefit. HPV particles mainly bind to the ECM on syndecan-1 HS chains. This attachment allows the setting up of the viral entry step in both in vitro and in vivo models. For the late steps of virus secretion, the disassembly and cleavage of the HS chains enhance infectious-HPV virus release from infected cells [77]. Heparanase-1 inhibitors dramatically reduce HPV virus release, providing evidence of the essential role of this enzyme in the HPV lifecycle and pathogenesis [77]. Infectivity assays with HPV-16 particles showed that OGT2115 (20 µM) significantly reduced (by more than 60%) the ability of the ECM-bound virus to infect the keratinocytes (HaCaT cell line) [77]. These findings may suggest additional targets for preventing HPV infection.

### 4.3. Dengue Virus (DENV)

Heparanase-1 is also involved in the infection cycle of RNA viruses, such as the positive single stranded RNA virus DENV, causing the most prevalent mosquito-borne disease in humans, which is a major public health issue worldwide. Dengue severity depends on several determinants, in particular, the escape of fluids and molecules from the bloodstream into tissues, which can lead to fatal complications. Dengue virus is able, through its NS1 protein, to induce the hyperpermeability of endothelial cells and systemic vascular leakage in vitro and in vivo. DENV disrupts the endothelial glycocalyx layer on human pulmonary microvascular endothelial cells by inducing heparanase-1 activity via cathepsin L activation. Specific heparanase-1 inhibitors may prevent the pathologic effect of the DENV NS1 protein on endothelial hyperpermeability. OGT2115 (1 µM) prevented both the disruption of the endothelial glycocalyx layer and the hyperpermeability of human pulmonary microvascular endothelial cells (HPMEC) triggered by dengue virus (DENV) [76], implying that heparanase-1 inhibition may interfere with viral pathogenesis.

### 4.4. Hepatitis C Virus (HCV)

We have recently documented the involvement of heparanse-1 in HCV infection [79], where it plays a proviral role. In this study, heparanase-1 expression was found to be induced in a NF-κB dependent manner, in vivo, in liver biopsies of HCV infected patients and, in vitro, in cultured hepatocytes. These results were supported by the restoration of the basal heparanase-1 expression after clearing the infection with the HCV antiviral drug, sofosbuvir. We also showed that heparanase-1 had no effect on HCV’s attachment or entry step, nor on viral RNA replication. In contrast, it promoted HCV egress and particle release from the infected cells through CD63 exosome secretion. This study emphasized the indispensable role of heparanase-1 in the HCV lifecycle as HCV infection was strongly reduced in heparanase-1 deficient cell lines or by using a specific inhibitor. In fact, we have blocked the enzyme with the HS mimetic muparfostat (PI-88) and observed a dose-dependent inhibition of HCV replication with a half-maximal inhibitory concentration (IC_50_) of 100 µg/mL. We also observed a significant inhibition of HCV infection with OGT2115 with an IC_50_ of 5 µM (data not shown).

### 4.5. Severe Acute Respiratory Syndrome Coronavirus 2 (SARS-CoV-2)

The COVID-19 pandemic from 2019 inspired many laboratories from all over the world to search for a remedy. As for other positive strand RNA viruses, HS are essential for the attachment step of SARS-CoV-2. Moreover, HS was identified as a co-receptor of the angiotensin-converting enzyme 2 (ACE2) for the entry step of SARS-CoV-2 in human cells [62]. In this respect, heparin, which is a heparanase-1 acting compound, has been tested as an antiviral. Treatment with it resulted in an inhibition of the SARS-CoV-2 invasion of Vero cells by up to 80% at concentrations from 6.25 to 200 μg/mL [80]. The doses are achievable through prophylaxis and within the range deliverable by nebulization. The inhibition of the viral infection was due to an overlap between the binding sites of heparin/HS on spike (S1) protein receptor-binding domain (S1 RBD) and that of ACE-2 [80]. Similarly, the synthetic HS mimetic pixatimod (PG545) was reported to bind to and destabilize the S1-RBD, which prevented its binding to ACE2 [62]. Consequently, assays with multiple clinical SARS-CoV-2 isolates showed that pixatimod potently inhibited the infection of Vero E6 and human bronchial epithelial cells at concentrations within its safe therapeutic dose range (half-maximal effective concentration (EC_50_) of 2.4–13.8 µg/mL). Furthermore, when pixatimod was given as a single prophylactic treatment of 16 mg/kg one day prior the infection of K18-hACE2 mice, it resulted in an attenuation of SARS-CoV-2 viral titer and COVID-19-like symptoms [62]. A very recent report showed that the chemically modified (100% N-acetylated and glycol-split) heparin roneparstat (SST001) significantly decreased the infectivity in vitro of SARS-CoV-1 and SARS-CoV-2 by suppressing the S protein-mediated entry [81]. The investigators used VSV-SARS-CoV-1 or VSV-SARS-CoV-2 chimeric viruses (VSV-eGFP-SARS-CoV-1, VSV-eGFP-SARS-CoV-2), in which the VSV G gene was replaced by the SARS-CoV-1 (or SARS-CoV-2) S gene. Treatment with roneparstat resulted in a dose-dependent decrease in infected Vero-E6 cells, with an IC_50_ of 0.05 mg/mL. Roneparstat inhibited the SARS-CoV-2 infection of Vero-E6 cells, with IC_50_ of 0.07 mg/mL. In the macrophages isolated from SARS-CoV-2 infected patients, an elevated heparanase-1 level was found and it correlated with greater rates of pro-inflammatory cytokines (IL-1β, IL-6, TNF-α and CCL2) compared to the healthy controls. Eventually, the authors of this study showed that roneparstat treatment could dampen this proinflammatory immune response mediated by the macrophages and, thus, it may serve as a dual-targeted therapy to reduce viral infection and inflammation in COVID-19 [81].

### 4.6. Porcine Reproductive and Respiratory Syndrome Virus (PPRSV)

In addition to the involvement of heparanase-1 in human virus-related pathologies, this enzyme is also tightly linked to animal pathogens, such as PRRSV, which may cause considerable economic losses in the pork industry. PRRSV also uses HS for its attachment to cell surface target cells. PRRSV was shown to upregulate heparanase-1 expression via NF-κB activation, leading to HS degradation. This activation of heparanase-1 is also supported by the fact that the cysteine protease cathepsin L is also upregulated by PPRSV. In fact, cathepsin L cleaves proheparanase-1 in endosomes to obtain mature and enzymatically active heparanase-1. The activated heparanase-1 then cleaves HS, resulting in PRRSV virion release [78]. Although the authors did show an effect of heparanase-1 exogenous overexpression or downregulation on virus infection, they did not use any pharmacological approach to block heparanase-1 activity. However, their data strongly suggest that heparanase might be an excellent candidate for the development of future antiviral strategies against PRRSV infection. This hypothesis should be investigated in future work.

## 5. Heparanase-1 Inhibitors as Future Antivirals?

The many virus examples described in this review provide a rationale for considering heparanase-1 as good target to inhibit one or several steps of virus lifecycles or virus-associated pathogenesis, including: attachment and entry; virion release or pro-inflammatory cytokine production. The above-mentioned examples clearly point out a potential niche for heparanase-1 inhibitors in treating virus infections. More experimental data are needed to show whether heparanase-1 is implicated in other viral infections with a huge health, social and economic burden, such as human immunodeficiency virus type 1 (HIV-1) and hepatitis B virus (HBV). In addition, clinical studies are necessary to better characterize the safety of some compounds, such as OGT2115 and other molecules that are being developed. Heparanase-1 inhibitors may be considered as dual-acting drugs in cancer treatment and in viral infections and could be used, in theory, for cancer patients with acute or chronic viral infections. Eventually, the development of novel class antivirals targeting cellular protein could alleviate the resistance problems seen with some current antiretroviral therapies.

## Figures and Tables

**Figure 1 viruses-15-00237-f001:**
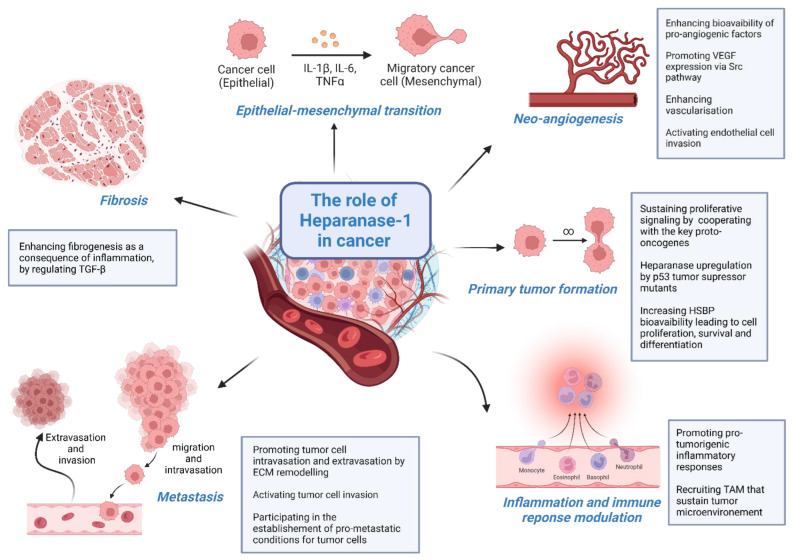
Major contributions of heparanase-1 in cancer. The enzymatic and non-enzymatic activities of heparanase-1 are involved in carcinogenesis, from the very early steps of tumor initiation to late stages including metastasis and invasion. Heparanase-1 plays a major role in several processes e.g., ECM remodeling, HS turnover, cell migration, tissue vascularization and remodeling, inflammation, and proliferation. As a global consequence, heparanase-1 contributes significantly to the establishment of the tumor microenvironment. Abbreviations: ECM (extracellular matrix), HSBP (heparan sulphate binding proteins), TAM (tumor-associated macrophages). [Created with BioRender.com].

**Figure 2 viruses-15-00237-f002:**
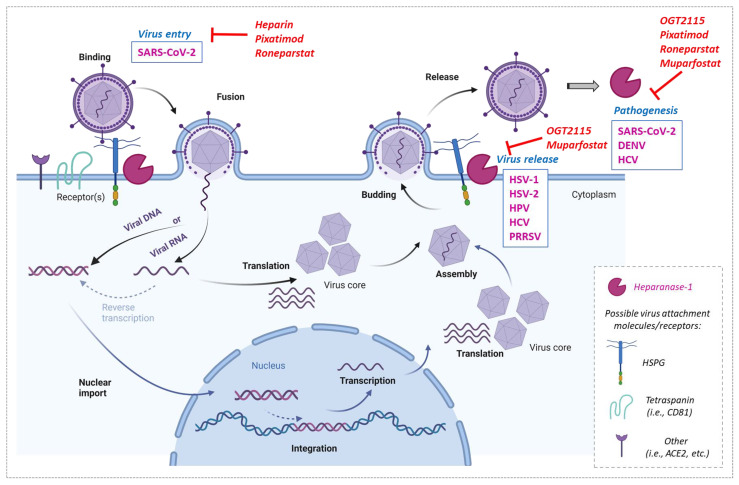
Impact of heparanase-1 on virus life cycles and their inhibition. Possible steps of the life cycle of RNA or DNA viruses for which heparanase-1 is thought to be involved are shown: binding and fusion. For an RNA virus without nuclear passage: translation and synthesis of viral proteins and genome, assembly of viral components, budding, and release of progeny virions. For an RNA virus with nuclear passage or a DNA virus: nuclear import (preceded by provirus formation in the case of retroviruses), integration (optional), transcription and synthesis of viral mRNAs, export and translation of viral mRNAs into viral proteins, assembly, budding, and release. Heparanase-1 is implicated in virus entry, release and/or pathogenesis when overexpressed. Inhibition of heparanase-1 with chemical compounds could block entry, release and/or pathogenesis of different viruses that require heparanase-1 during infection. [Created with BioRender.com].

**Table 1 viruses-15-00237-t001:** Role of heparanase-1 in viral infections.

Viruses	Role of Heparanase-1 in Infection/Pathogenesis	Heparanase Inhibitors Blocking Infection	References
Herpes simplex virus (HSV-1)	Increased virus release. ECM damage leading to disease pathologies	OGT2115	[73,74]
Herpes simplex virus (HSV-2)	Increased virus release	OGT2115 Pixatimod (PG545)	[75]
Dengue virus (DENV)	Increase in severity of disease symptoms	OGT2115	[76]
Human papilloma virus (HPV)	Increased virus release	OGT2115	[77]
Porcine respiratory and reproductive syncytial virus (PRRSV)	Increased virus release	N/A	[78]
Hepatitis C virus (HCV)	Increased rate of infection/virus release	Muparfostat (PI-88)	[79]
SARS-CoV-2	Attachment, entry	Heparin Pixatimod (PG545) Roneparstat (SST001)	[62,80,81]

## Data Availability

Not applicable.

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
