# Peer review of "Heparanase-1: From Cancer Biology to a Future Antiviral Target"

_viruses, 2023, doi:10.3390/v15010237_

Round 1

Reviewer 1 Report

This review manuscript provides an outline of Heparanase-1, an enzyme playing a role in cancer which also participating in infection of numerous viruses could be a promising target for antiviral drugs. The authors raised an interesting focus to have an alternative target for antiviral drugs against a broad spectrum of viruses which is necessary for preventing a new pandemic . While this is a potential target, overall I found practical difficulties of applying to specifically viruses, as well as lacking critical evaluation of the literature for this perspective.  Thus, major revisions are needed to make this an impactful contribution to the field. The concerns are outlined below: 

Major Points:

  1. Section 3, no attempt was made to explain the  mechanism of action of these inhibitors in cancer therapy specifically. The well-described in this section would help to understand their potential use for antiviral drugs. 

  2. My main concern of the review is  the lack of discussion and citation of heparanase -1 inhibitors to prevent virus replication to raise the main focus of the manuscript,  especially to the RSV and PRRSV part.

  3. Table 4, I would recommend adding one column with its possible inhibitors to create a greater informative table, for example, which heparases-1 inhibitors exhibited antiviral activity against HSV-1.  Additionally, change the references into numbers ( Agelidis et al., 2017 [77] to [77]. Furthermore, the table should come after section 4. 

  4. Recent references related to manuscripts are neglected. Additionally, please carefully check the citation of correct references. For example, I did not find reference [78] in section 4.1. 

Minor Points:

  1. In section 2, please break this section into multiple paragraphs with the specific role of heparases-1 in cancer which were well-described in Figure 1.

  2. The introduction of Heparanase -1 in the 2nd paragraph of section 1 and the 1st part of section 2 made the feeling of repeated discussion. 

  3. Please keep all abbreviations throughout the manuscript, e.g .  HS for heparan sulphate, HSV-1 for herpes simplex virus 1, etc

  4.  Figure 2: step (3) should be distinguished by different numbers between cytoplasm viruses and nuclear viruses. Additionally, roneparstat was stated in the figure 2 but there was no mention about the inhibition against any described viruses in section 4. 

  5. Please remove the space between two words, ex: line 50, 114, 174, etc.

  6.  Section 4.7: I assume it should be PRRSV not RSV. 

Reviewer 2 Report

This is a generally well written review of two potential uses for the therapeutic use of regulation of Heparanase-1.  I believe it would be of interest to the readers of viruses. I just had a few comments mostly in light of clarification.

The title of the manuscript might need to be re-thought.  It suggests to me that your discussion of cancer treatments would lead to some insight into how they might be used as antivirals?  However I more kept getting the impression the two topics were almost treated as separate phenomena?  Either make the link stronger or maybe re-work the title.

I would move the table down a bit as it seems to precede the first reference to it?

Figure 2 is nicely done, but doesn't provide much detail of the proposed mechanisms.  Include a bit more identification for what the molecules presented in the membrane are?  The legend only identifies Heparanse-1 

Minor suggestions:

line 117 - remove "as stated earlier"  doesn't really add anything

line194 - This first sentence needs to be reworded.

line 289 - needs re-wording.  I don't think the viruses were concerned.

Round 2

Reviewer 1 Report

The authors have addressed and revised almost all of my points which enhanced the comprehension of the manuscript. I only have one more comment. In section 2, the authors have addressed the removal of RSV viruses, so should the rest of manuscripts.  Please check again, eg, figure 2 , line 226,